# A Highly Interactive Honeypot-Based Approach to Network Threat Management

**Xingyuan Yang [1,2], Jie Yuan [1,2,*], Hao Yang [1], Ya Kong [1] , Hao Zhang [1] and Jinyu Zhao [1]**

1 School of Cyberspace Security, Beijing University of Posts and Telecommunications, Beijing 102206, China
2 Key Laboratory of Trustworthy Distributed Computing and Service (BUPT), Ministry of Education, Beijing 100876, China
* Correspondence: yuanjie@bupt.edu.cn; Tel.: +86-13810019079

**Abstract:** In this paper, considering the problem that the common defensive means in the current cyber confrontation often fall into disadvantage, honeypot technology is adopted to turn reactive into proactive to deal with the increasingly serious cyberspace security problem. We address the issue of common defensive measures in current cyber confrontations that frequently lead to disadvantages. To tackle the progressively severe cyberspace security problem, we propose the adoption of honeypot technology to shift from a reactive to a proactive approach. This system uses honeypot technology for active defense, tempting attackers into a predetermined sandbox to observe the attacker's behavior and attack methods to better protect equipment and information security. During the research, it was found that due to the singularity of traditional honeypots and the limitations of low-interactivity honeypots, the application of honeypot technology has difficulty in achieving the desired protective effect. Therefore, the system adopts a highly interactive honeypot and a modular design idea to distinguish the honeypot environment from the central node of data processing, so that the honeypot can obtain more sufficient information and the honeypot technology can be used more easily. By managing honeypots at the central node, i.e., adding, deleting, and modifying honeypots and other operations, it is easy to maintain and upgrade the system, while reducing the difficulty of using honeypots. The high-interactivity honeypot technology not only attracts attackers into pre-set sandboxes to observe their behavior and attack methods, but also performs a variety of advanced functions, such as network threat analysis, virtualization, vulnerability perception, tracing reinforcement, and camouflage detection. We have conducted a large number of experimental comparisons and proven that our method has significant advantages compared to traditional honeypot technology and provides detailed data support. Our research provides new ideas and effective methods for network security protection.

**Keywords:** high interaction honeypot; cyber threat analysis; virtualization; vulnerability detection; traceability reinforcement; masquerade detection

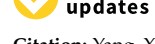



## 1. Introduction

The development of artificial intelligence, the Internet of Things, computers and Internet technology is transforming diverse facets of human society. Simultaneously, fresh information security and network security threats are constantly emerging, and the potentially devastating effects are increasingly prominent. There is a serious asymmetry in network attack and defense. Traditional information security and network security technologies, such as cryptography, firewalls, intrusion detection, and authentication, all share a common characteristic: passive defensiveness. These technologies adopt passive security strategies and cannot make an instant response to the attacker's behavior. The evolution of attack behavior in recent times has rendered traditional technologies incapable of preemptively detecting and preventing network attacks. Instead, these technologies can only reproduce the attack behavior after the attack has occurred, thereby merely

supplementing existing defense measures [1]. Such an approach falls short of meeting the current demands of network protection. Honeypot technology serves as an active defense strategy that can effectively supplement the limitations of traditional network protection measures.

The concept of honeypots was first introduced in 1991 by Clifford Stoll in his book "The Cuckoo's Egg". At that time, only some professional network security personnel would trace the attackers through some hosts that were actually attacked. Starting in 1998, honeypot technology attracted a group of developers who developed a series of honeypot tools, such as DTK developed by Fred Cohen, and Honeyd developed by Niels Provos. Honeypot technology, which rose in 1999, aims to convert passive network defense into active defense. Through active interaction with intruders, users can gain valuable insights into their motives, intentions, and means. This approach offers a technical solution for predicting network security trends and identifying potential threats before a formal attack occurs. By doing so, it can help mitigate the harm caused to network infrastructure by malicious attack behavior. Nowadays, honeypot technology has only been born for more than 20 years, but its maturity and popularity have reached a very high level domestically and internationally.

According to Lance Spitzner's definition, the honeypot is an information system resource whose value lies in being used unauthenticated and illegally. In other words, a honeypot is a deliberately crafted trap that can be created in either a virtual or real network environment. Its purpose is to lure attackers, identify malevolent activities, and gather data on attack tactics, tools, and objectives. The information gathered from honeypots is used to analyze the vulnerabilities of both attackers and the systems being imitated. This analysis indirectly facilitates the protection of the actual network environment [2].

Our research focuses on current honeypot technology and we have found that high-interaction honeypots, as discussed in "The difference between high-interaction honeypots and low-interaction honeypots" and "Design and implementation of highly interactive honeypots", are more effective at deceiving attackers and attracting their attention. These honeypots simulate realistic FTP and HTTP services and can create 'baited' files to gather in-depth information on hacker behavior. However, if a highly skilled hacker successfully infiltrates the site and takes control of the system, it will completely expose the network environment that should be protected, which poses a huge security risk. The article "Application and research of honeypot technology in information security defense" simply describes several core modules of honeypots but does not specify the implementation principle of honeypot application, which is not conducive to the wide application of honeypot technology. In this study, we present a detailed overview of our designed honeypot management system model. This model is designed to enhance the ease of use of the system for specific purposes.

We conducted extensive research on the popular open-source honeypot products available in the market, including Hfish, SSH-mitm, and Shadow Daemon. Our analysis revealed that each of these products has certain limitations.

In terms of interactivity, Shadow Daemon and SSH-mitm are categorized as high-interactivity honeypots, which simulate genuine operating environments and significantly enhance the attractiveness and concealment of the honeypots for potential attackers. High-interactivity honeypots offer more comprehensive and detailed data, but their susceptibility to being compromised by attackers creates a high risk of losing the ability to record attack information in the event of a breach. The low-interactivity honeypot Hfish lacks the ability to simulate a real operating environment, resulting in low concealment and attractiveness compared to high-interactivity honeypots such as Shadow Daemon and SSH-mitm. As a consequence, the Hfish honeypot is prone to collecting misleading or incomplete attack data. However, due to its low interactivity, the logging system is relatively separated from the honey bait and cannot be easily breached by attackers to collect data.

In terms of simplicity, honeypots such as Hfish, SSH-mitm, Shadow Daemon, Dionaea, etc. are all managed through the command line. Requiring concise commands (e.g.,

shutdown or disconnect) to be issued to perform specific actions, the management of the honeypot can be challenging, rendering it unsuitable for large-scale deployment. Secondly, the lack of a graphical interface also prevents the data from being represented visually, which is not conducive to the user's analysis of the attack data. In contrast, honeypots such as Honeyd employ a graphical user interface for configuration and display, enabling the emulation of various systems and services with customized virtual host configurations. This greatly reduces the difficulty for users to operate and analyze.

In terms of versatility, honeypots such as Honeyd offer versatility by supporting multiple operating systems, including Linux, Unix, and Windows systems. On the other hand, honeypots such as Kippo and Dionaea can only run on Linux operating systems, making their versatility relatively narrow.

In terms of information collection, Honeyd and Hfish [3] can only collect the attacker's IP address and port information, with relatively small amounts of information collected; honeypots such as Kippo and Dionaea can capture the attacker's IP address, login information, and command inputs. Furthermore, Kippo has the ability to analyze collected attack information to provide users with valuable insights.

Our research on the aforementioned products indicates that traditional low-interaction honeypot systems struggle to handle the growing complexity of network environments. Conversely, our proposed high-interaction honeypot system provides greater interactivity by simulating open and realistic operating systems and services. This approach confuses attackers and enables the capture of realistic attack data. To enhance network security, honeypots are encapsulated using modular technology, and a modular and high-interaction honeypot threat management system is designed with the integration of various honeypots.

After a large number of comparative experiments, it was found that our method has significant advantages compared to traditional network threat management methods: As the core module, the central node implements various functions such as data storage, deployment of multiple honeypots, connection control, and data processing. Through performing these operations on the honeypots, data is collected, organized, summarized, and analyzed to generate event reports and attacker profiles. This facilitates easy viewing and usage of attack information templates.

The proposed method for constructing a highly interactive honeypot system involves using multiple highly interactive honeypots as tools for information sniffing. These honeypots are constructed to provide specific services by setting up honey baits partially constructed with real information to entice attackers to carry out attacks. By doing so, information about attackers' attack behaviors and methods can be collected. The system is designed to be modular and highly interactive, encapsulating various honeypots. The central node, serving as the core module, implements data storage, deploys multiple honeypots, controls connections, and performs a series of data processing functions. Various operations are performed on the honeypots to collect data, organize, summarize, and analyze the data, forming event reports and attacker profiles for easy viewing and use of attack information templates.

Using third-party tools as an information transmission channel, the data collected by the honeypot is encrypted and reliably transmitted to the central node. The central node processes this data by deploying multiple honeypots, controlling connections, and organizing, summarizing, and analyzing the data. The result is the creation of event reports and attacker profiles that can be easily viewed and used with attack information templates. The data report and attacker profile formed by the central node are then transmitted to the user's front-end interface. A concise front-end user interface serves as a window for system-user interaction and allows users to utilize the honeypot threat management system without needing relevant professional knowledge.

Section 2 will introduce the current cyber threat management methods and examine the benefits and drawbacks of different cyber threat management approaches in various domains, including how they can support our proposed cyber threat management approach. In Sections 3 and 4, we will elaborate on our network threat management approach and

its implementation specifics. In Section 5, we will expound on the actual operational effectiveness of our proposed network threat management approach in this paper. In other parts of the paper, the detailed structure of the high-interactivity honeypot threat management platform will be explained in detail, as well as its availability and potential.

## 2. Survey of Existing Network Threat Management Methods

Before introducing the proposed network threat management method in this paper, it is necessary to compare and introduce the mainstream network threat management methods available in the market. This will help readers to understand the different network threat management methods, and their functions, and highlight the advantages and disadvantages of various network threat management methods. It is important to understand the principle construction of these methods in order to gain a better understanding of the proposed method. At the same time, studying other network threat management methods, can also further help the authors to improve our network threat management method.

Currently, the main methods for managing network threats include intrusion detection systems, intrusion prevention systems, vulnerability scanners, log auditing systems, identity authentication, access control, honeypot protection, etc: Intrusion Detection System (IDS) is designed to detect and identify unauthorized access or attack activities. It is usually used to monitor network traffic, recognize malicious activities, and alert network administrators [4]. The main types of IDS include snort, Suricata, etc. IDS uses two main technologies: signature technology and behavior analysis technology. Signature technology identifies attacks by matching known malicious code signatures. Behavior analysis technology recognizes malicious activities by analyzing network and system behavior [5].

The intrusion prevention system (IPS) aims to detect and prevent unauthorized access and attacks. IPS is similar to an intrusion detection system (IDS), but IPS also has the ability to prevent attacks. IPS uses a rule engine that uses rules to detect specific network attack patterns and block them when detected. An intrusion defense policy is established and predefined policies are utilized to detect and prevent specific types of attacks. A signature database is formed that contains the digital signatures of known attacks for quick identification of known attacks. Network pattern analysis is performed by using algorithms to identify anomalous network patterns and block them when an attack is detected. IPS can be installed at the network edge for real-time monitoring of network traffic or on specific security devices to protect the network from attack. IPS is commonly used to prevent hacker attacks such as denial of service (DoS), viruses, malware, and network probing. It can effectively prevent these attacks and ensure the security and reliability of the network.

A vulnerability scanner is an automated tool used to identify security vulnerabilities in a network. It scans the network and systems to identify weaknesses that may be vulnerable to attack and provides information to administrators so they can take action to fix the vulnerability. The vulnerability scanner detects the devices and services on the target network by performing a network scan. Then, it compares the results to a database of known vulnerabilities and runs vulnerability checks. If the scanner identifies a potential vulnerability, it generates a detailed report with recommendations for repairing the vulnerability and sends it to the administrator.

A log auditing system is a tool used to record and analyze information about activities on the network. It helps administrators identify unusual activity, assess network security risks, and prevent malicious attacks and data breaches. The system collects detailed activity information, such as user logins, file access, and network traffic, from various devices on the network. The collected activity information is stored on a log server and analyzed using rules defined by the administrator. The system can provide visualized reports to help administrators identify abnormal activities, such as frequent login failures or malware activity. If any threat activities are detected, the log auditing system can send an alert to the administrator and automatically trigger corresponding preventive measures.

Authentication and access control are two important parts of network security. Their purpose is to protect network resources and prevent unauthorized users from accessing

sensitive information. Authentication is the process of determining a user's identity. It ensures that only users who have been authenticated can access network resources. Common authentication methods include username/password, digital certificates, biometrics, etc. Access control is the process of controlling a user's access to network resources based on their identity and other factors (such as user group, time, etc.).

Next, we will conduct a detailed analysis and comparison of several network threat management methods from various perspectives:

*2.1. Interactivity Comparison*

The interactivity of a network threat management system refers to the interaction mode and the breadth of information acquisition between the system and the security object. This includes the information acquisition method of the system, operation methods, degree of information acquisition, and the interaction mode between the user and the system, such as interface design, report generation, data visualization, etc. Good interactivity can help users more easily understand and use the system, improve the efficiency and effectiveness of the system, and also help the user to obtain more detailed information about the security object, further enhancing the reliability of the system.

Compared to other network threat management methods, the interactivity of low to medium-interactivity honeypots is lower. For example, Dionaea and Cowrie: Dionaea is an open-source low to a medium interactivity honeypot system. It aims to simulate common network services to collect information about potential attackers and supports multiple protocols, including HTTP, FTP, SMTP, etc. Dionaea can collect information such as the remote host's IP address and location, attempted login username and password, connection attempt time and frequency, as well as uploaded or downloaded files. Cowrie is an open-source low to medium interactivity honeypot system specifically designed for collecting and analyzing information about SSH and Telnet attackers. It mimics a real SSH and Telnet server to collect information about the attacker. The information collected by Cowrie includes the remote host IP address and location, login attempt username and password, connection attempt time and frequency, and content of command line input. However, both honeypots use a command-line interface for interaction with users, greatly hindering interaction efficiency between users and the system. Additionally, both honeypots can only collect basic intrusion information, and more detailed information about the intruder cannot be collected, causing great obstacles for network analysis. Therefore, in terms of interactivity, low- to medium-interactivity honeypots are undoubtedly poor.

Compared to the low interactivity of low to medium interactivity honeypots, intrusion detection systems and log audit systems have relatively higher interactivity. As an example of a highly representative intrusion detection system, Snort uses a rule-based method to detect network intrusions and security threats. Administrators can write custom rules to detect specific types of malicious traffic, or they can use pre-defined rules provided by Snort or third parties. Snort can run in real-time and alert administrators to suspicious activity, or it can be used for offline analysis of network traffic logs. Compared to low-interactivity honeypots, Snort is more accurate in its judgment of intrusion behavior due to its ability for users to write their own rules and describe and review the characteristics of intrusion behavior. The information collected by Snort is diverse and comprehensive, including network attacks such as denial of service (DoS), buffer overflows, and SQL injection attacks. It can also detect malware and other types of malicious software transmitted over the network, as well as policy violation behavior, such as unauthorized access to restricted network resources [6]. However, Snort requires users to define intrusion behavior in advance in the form of a command line, and the logs of intrusion response are also displayed in the form of a command line, which creates great difficulties for the interaction between the system and the user. Taking Graylog as an example, Graylog is an open-source log management system that provides log collection, storage, search, and analysis functions. It enables centralized log management of multiple data sources and supports real-time processing and analysis of data. Graylog uses Elasticsearch as the storage engine and

MongoDB as the metadata storage. It provides a web interface that allows users to search and analyze log data and customize rules to monitor log messages and generate alerts. Graylog also supports the visualization of log data, including charts and dashboards. It also offers a flexible plugin architecture that can extend its functionality and integrate with other tools. Overall, Graylog is a powerful log auditing system that provides an effective way for enterprises to centrally manage and analyze log data. However, like Snort, Graylog also requires users to set up the log auditing mechanism in advance and uses a command-line form for user–system interaction. Additionally, the defense mechanisms of both Snort and Graylog can only react after the intrusion has taken place, and they have a lag in protecting security objects.

In response to the relative passiveness of intrusion detection systems, intrusion prevention systems (IPS) were developed. IPS is similar to intrusion detection systems (IDS), but also has the ability to prevent attacks. Here we take the Controller Area Network Intrusion Prevention System Leveraging Fault Recovery method proposed by Habeeb Olufowobi, Sena Hounsinou, and Gedare Bloom as an example. This is a new type of IPS for the CAN bus that can prevent remote message injection attacks and trigger remote damaged ECU based on reboot recovery. This method combines CAN and IDS and measures their ability to detect attacks in the delay within the bus speed. The detector node can decide the message frame that is broadcast between the last transmission of the arbitration field and the end of the message frame. Its actual recovery period can reach 20 milliseconds. The recovery mechanism converts the abnormal node to a bus shutdown state in about 6 milliseconds, which is less than the period of a valid message frame. This IPS can be integrated into real car systems, mitigating the impact of attacks while identifying them and measuring the performance degradation of reboot-based recovery, and preventing future attacks [7]. In terms of interactivity, this method offers several advantages. Firstly, it provides detailed information on the intrusion detection system. This level of detail enables users to have a more comprehensive understanding of the system's performance and any potential security threats. Additionally, the method has the ability to predict potential intrusions, which allows for more proactive security measures. Moreover, this approach includes a graphical user interface, which enhances the user–system interaction experience. The interface provides users with an intuitive and easy-to-use platform to interact with the system. This convenience can improve overall system usability and promote user adoption.

Identity authentication and access control are currently the most commonly used network threat management methods, widely used in systems such as Windows, Linux, and Unix-like systems. This method has undergone significant market improvements and now features a robust graphical user interface with a user-friendly design, which enhances user–system interaction. Specifically, the method typically includes a graphical user interface with 280 different graphical elements that make it easier for users to navigate and interact with the system. Notwithstanding its benefits, this method has a significant drawback. Specifically, it employs a passive protection approach that does not actively monitor security objects or record intrusion behavior. As a result, any intrusion information is left for security personnel to refer to after the fact, rather than being detected and prevented in real-time. There is a serious lack of interaction between the system and security objects.

Compared to previous network threat management methods, which interact passively with attack behaviors, vulnerability scanners offer a proactive approach to protecting networks by actively interacting with security objects. Here we take MoScan as an example, it is a model-based scanner for detecting security vulnerabilities in SSO implementations. It tests the protocol participants as a black box but builds a state machine to represent the logic of the login process and guide the generation of test cases. This vulnerability scanner can scan the security object for vulnerabilities based on its own vulnerability database before an attack occurs, and the scan information is detailed and has a high degree of interactivity with the security object [8]. The scanner has been improved by the authors of the paper, with a graphical user interface, making the interaction between the system

and the user high. Nonetheless, in the event of an actual intrusion by a malicious actor, a vulnerability scanner would be rendered ineffective. It would be incapable of providing any meaningful response, documenting intrusion details, or identifying previously unknown vulnerabilities or targeted attacks. Its capabilities would be limited to scanning for known vulnerabilities that exist within its vulnerability database.

In conclusion, in terms of interactivity, authentication and access control exhibit the lowest level of interactivity, whereas the interaction between honeypots and intrusion detection systems is relatively poor. On the other hand, vulnerability scanners and intrusion prevention systems demonstrate more prominent interactivity, with each system emphasizing distinct areas.

*2.2. Tempting Comparison*

The temptation index of a Network Threat Management System pertains to the specific targets established by the platform through technical means. These targets are designed to facilitate the attackers in identifying and recognizing valuable assets, thereby making them more susceptible to attacks. The main manifestation is attack deception defense technology, such as honeypots, honeynets, honey traps, honey markers, lures, deception defense platforms, mobile defense targets (MTD), and mimicry defense, etc. [9]. The technology aims to protect the organization's network, system, and application assets with the goal of security protection. It realizes the goals of obtaining attacker information, increasing attack difficulty, and sticky defense through the exposure of facts, hiding facts, exposing lies, and hiding lies, etc. It is a part of active defense and can use deception defense technology to build a trapping, monitoring, and tracing system. The establishment of tempting targets can effectively increase the cost of an attack, ensnare attackers by presenting them with counterfeit devices or services, and encourage the adoption of erroneous attack methodologies and tools. This approach represents a specific application of attack deception defense.

Access control and authentication systems are designed to protect the confidentiality and non-repudiation of data. Based on authentication, access control technologies are applied to regulate the access scope of users or programs to resources and systems, ensuring that authorized users can use the system and data legally, and realizing controlled management of resources. At the same time, users cannot operate information outside of their own permission range. Due to the different design intentions, this system does not consider deception attacks, so it does not have enticement, but it can complement the network threat management system with deception defense technology, forming a multi-layered protection system.

The intrusion detection system and log audit system detect intrusions or attempts on the system by inspecting network behavior, network logs, audit data, and other network information. They take action to block the intrusion based on preset rules. For example, the intrusion detection system in Qingteng Cloud Security has a multi-dimensional intrusion-sensing network and its core functions are divided into four modules: intrusion handling, real-time event monitoring, intrusion analysis, and real-time intrusion warnings. Different detection methods are used to respond to both ongoing and past intrusions. Graylog is an open-source log management system that receives logs through different input interfaces and provides a web access interface for users. It uses Elasticsearch to index and store logs and MongoDB to store configuration information. Log information enters Graylog through Inputs, passes through Extractors if log field extraction is set, then arrives in the Streams log stream, and finally is written to the specified Index based on the Streams log stream settings. Alarms are triggered based on the matching information set in the log stream. Similar to access control and authentication systems, the design purpose of these two systems is to promptly detect intrusions in the network environment and take countermeasures. Therefore, a single intrusion detection system or log audit system does not have the same level of appeal. However, its handling of intrusion information can effectively process information obtained from attacker deception defense techniques, increase defense reserves, and better respond to network attacks.



An intrusion prevention system is a computer network security facility that supplements anti-virus software and firewalls. The network device under consideration functions as a monitoring mechanism that actively monitors the behavior of data transfer across networks or network devices. In the event of any identified abnormal or harmful data transfer behaviors, this device can swiftly interrupt, adjust, or isolate such actions to prevent potential damage or compromise. A vulnerability scanner is a hardware device used to scan vulnerabilities in an enterprise network. Traditional vulnerability scanners have a program inside the hardware that automatically detects weak points in remote or local host security. However, the vulnerability scanner provided by QualysGuard based on the SaaS cloud service architecture is simply a tool for collecting various system situation data for comparison with vulnerability data in the Qualys terminal SOC to confirm the system status. Due to the original intention of the design, both intrusion prevention systems and vulnerability scanners are not attractive.

Compared to several of the aforementioned network threat management methods, medium- and low-interactivity honeypots are somewhat tempting [1]. In the same vein, Dionaea and Cowrie are used here as examples. Dionaea is a honeypot that has been designed to be minimally interactive, allowing attackers to target simulated systems and services that do not represent real product systems. The purpose of the honeypot is to expose the vulnerabilities and targets that attackers may exploit, without risking the security of actual systems. This design makes the honeypot easy to install and configure, and the honeypot system has almost no security risks, but imperfect simulations can reduce the ability to capture data and be easily recognized by attackers and lose effectiveness when faced with increasingly mature honeypot recognition technology. Cowrie, as an open-source low-to-medium interactive honeypot system, mainly simulates a real SSH and Telnet server to collect information about attackers. Like Dionaea, this system also encounters a similar problem. While it may have some degree of allure to attackers, it can be readily detected with the advent of increasingly sophisticated honeypot identification technologies. As a result, the system fails to achieve its intended objectives and instead consumes valuable system resources, thereby inflating network defense costs. Compared with traditional defense technologies, medium- and low-interaction honeypots have a certain temptation, but they are gradually unable to meet the demand and are only suitable for specific environments.

High-interaction honeypots are similar to real systems in that they can be completely compromised, allowing the attacker to gain full access to the system and use it as a stepping stone to launch further network attacks. Although the deception of the honeypot is increased and better able to attract hackers, if a hacker with a higher technical capability invades the site and controls the system, it will completely expose the network environment that should be protected, posing a huge security risk.

In summary, traditional access control and authentication systems, intrusion detection systems, log auditing systems, intrusion prevention systems, and vulnerability scanners are not typically effective in utilizing deception defense techniques due to their lack of enticement. However, the technology and design principles employed in these systems can supplement deception defense techniques, enhancing the security of the boundary between revealing and concealing, thereby safeguarding target information and environments. While medium- and low-interaction honeypots may be initially tempting, their success rate will likely decrease over time due to advancements in honeypot identification technology. On the other hand, high-interaction honeypots can be more seductive as they utilize actual services to deceive; however, they also entail unpredictable potential risks. High-interactivity and low-medium-interactivity honeypots each have their own areas of focus, with high-interactivity honeypots generally offering greater value for study.

### 2.3. Security Comparison

The security of a network threat management system involves reinforcing network security protection measures to prevent breaches. This is achieved through technical means,

such as setting up a highly deceptive target that entices attackers to launch attacks, thereby obtaining valuable information about them. Through the arrangement of security targets, it can effectively increase the cost of attackers and induce them to adopt the wrong attack methods and tools, thus protecting the organization's network, systems, applications, and other assets. Security is defensive and plays an important role in the network security protection system. It not only captures attackers through decoy targets but also enables detailed analysis of attackers through monitoring, traceability and other technologies to provide strong support for later protection. By applying security technologies, it prevents and avoids network attacks, data leakage, and other security threats, thus ensuring the safety of data and information of security targets. Simultaneously, the integration of interactivity and security in a network threat management system is crucial. Interactivity pertains to the system's interaction with the user and the extent of information available to them. A well-designed interactive system enables users to use the system more easily, leading to improved efficiency and security. Additionally, it facilitates the acquisition of detailed security information, enhancing the system's reliability. As such, the security and interactivity of a cyber threat management system work in tandem to determine the system's security and efficiency.

There are many network threat management systems available in the market, including Honeyd, Dionaea, Kippo, Glastopf, Cowrie, Snort, etc. They have different implementations and are designed with different security architectures, which result in significant differences in terms of security. Honeyd is a small daemon that creates virtual hosts on a network. The hosts can be configured to run any service and can be adjusted to look like they are running specific operating systems. Honeyd allows a single host to declare multiple addresses for network simulation on the LAN. Honeyd improves network and self-security by providing threat detection and evaluation mechanisms. It also deters opponents by hiding real systems behind virtual systems. However, it is not designed with a CS architecture, which means that if the honeypot is compromised, the overall security of the system will be lost.

Furthermore, Dionaea is a honeypot designed to emulate a server in order to capture the attacker's data. It supports multiple protocols such as HTTP, FTP, TFTP, etc. The primary objective of this activity is to detect malware that exploits vulnerabilities within the network's service offerings. The ultimate aim is to obtain a copy of the malware for analysis. Given that software that provides network services may contain exploitable vulnerabilities, it is reasonable to assume that dionaea, which also provides network services, may have such vulnerabilities as well. To mitigate potential security risks, dionaea is configured to operate without high privileges and to run within a chroot environment. However, to execute specific operations that require elevated privileges, dionaea creates a child process at startup that is authorized to run those operations after the privileges have been granted and subsequently removed. This does not guarantee anything, but it should be harder to get root access to the system from an unprivileged user in a chroot environment. However, this just hides the weakness deeper, and a malicious attacker can still get special privileges through dionaea.

Kippo is a highly simulated low-interaction SSH honeypot, running as a daemon written in Twisted Conch, used to monitor data from attackers. Because it is a low-interaction honeypot, security is lacking. Furthermore, because it has fewer functions, the possibility of vulnerabilities in the system itself is relatively small, which increases Kippo's security in another dimension. Another attack point for Kippo is its daemon, which relies on Twisted Conch to write the daemon to self-reset and run, if the security of Twisted Conch is breached, the security of Kippo will also be affected.

Glastopf is a web application honeypot that simulates a web application to capture data from attackers. Web applications, databases, and cross-site scripting vulnerabilities create a substantial attack surface that can be exploited for a variety of malicious purposes, including website destruction, spam email propagation, website bot program creation, and drive-by download attacks. Glastopf is a low-interaction honeypot that mimics vulnerable web servers hosting many web pages and web applications with thousands of vulnerabilities.

Glastopf is easy to set up and receives thousands of attacks every day once it is indexed by search engines. Glastopf is a Python web application honeypot. Instead of vulnerability simulation, we use vulnerability-type simulation. Several common attack types are already established, including remote file inclusion through a built-in PHP sandbox, local file inclusion that provides access to files from a virtual file system, and HTML injection through POST requests. The security of Glastopf is not guaranteed due to its low interaction, but Glastopf has added a PHP sandbox to the security architecture, separating the program's runtime environment and program execution environment logic analysis, strengthening the overall security.

Cowrie is an SSH and Telnet honeypot used to monitor data from attackers. Cowrie is a medium–high interaction SSH and Telnet honeypot designed to record brute force attacks and shell interactions performed by attackers. In the medium interaction mode (shell), it simulates a UNIX system in Python, and in the high interaction mode (proxy), it acts as an SSH and Telnet proxy to observe the attacker's behavior towards another system. It can be run either as a simulated shell or as a proxy for SSH and Telnet to another system. It provides a complete and available Docker environment, which allows for deployment environment and physical environment isolation [10]. The security depends on the relatively secure Docker environment. As a medium-high interaction system, Cowrie has relatively high security.

Snort is a potent network intrusion detection system that surveils network traffic, identifying any attack behavior, and currently stands as the world's most significant open-source intrusion defense system (IPS). The Snort IPS uses a series of rules to help define malicious network activity and uses these rules to search for matching packets and generate alerts for users. Snort can also be deployed inline to block these packets. Snort has three main uses: as a packet sniffer (such as tcpdump), as a packet recorder—which is useful for network traffic debugging, or it can be used as a mature network intrusion defense system. Snort can be downloaded and configured for personal and business use. Unlike traditional honeypots, Snort has no disguise and relies on transparency to monitor real systems. From some perspectives, it has no security, it is merely an adjunct to the physical system and constantly checks for attack behavior.

These network threat management methods each have their own features and are suitable for different network security purposes, creating a diverse security landscape. Although honeypots are an effective network security tool, they cannot completely replace other security measures such as firewalls and encryption. Only by combining with other security technologies can the system's security be increased.

### 2.4. Simplicity Comparison

The simplicity of use of a network threat management system refers to the convenience of installing the system and the simplicity of operation for the user when performing network security activities. A system that is easy to use is more conducive to user operation and installation, thereby reducing the barriers to entry and having a significant impact on promoting and facilitating widespread usage.

As an illustration of an intrusion detection system, we can take the well-known example of Snort. Snort has three modes: sniffer mode, packet logger mode, and network intrusion detection system mode. The sniffer mode only captures network packets and displays them on the terminal, the packet logger mode stores the captured packets to disk, and the intrusion detection mode is the most complex, which can analyze packets, detect them based on rules, and respond accordingly. Snort has many functions, such as packet sniffing, packet analysis, packet detection, response processing, etc. Each module implements different functions, and each module is integrated with Snort as a plug-in. The system framework adopts a modular approach, which is not only simple and easy to understand, but also easy to expand functionality and convenient for users to maintain. However, since Snort adopts a custom rule method to detect network intrusion in the intrusion detection mode, although this increases the system's flexibility, the detection of

network intrusion behavior will be more accurate, the collected information is more diverse and comprehensive, but it also increases the cost of use and learning for Snort users. The command-line format used for rule definition and intrusion log function in this intrusion detection system somewhat undermines its simplicity.

The intrusion prevention system is exemplified by the Controller Area Network Intrusion Prevention System Leveraging Fault Recovery method. Its specific implementation mechanism is divided into four parts: data reconstruction, protocol recognition and protocol parsing, feature matching, and response processing. Prior to entering the IPS, the data undergoes IP fragmentation and TCP stream reconstruction to ensure the continuity of the application layer data. The system then identifies the application layer protocol by analyzing its content and decodes it in detail according to the specific protocol. Furthermore, the system performs a thorough packet feature extraction. Finally, the parsed packet features and signatures are matched, and if the signature is hit, a response processing is performed [7]. On the whole, the IPS is simple in terms of user-friendliness and is easier to expand. However, expanding the intrusion defense feature library also leads to an increase in the system's maintenance costs. At the same time, the increase in system functionality also increases the complexity of the system framework and the cost of maintenance and use.

The log auditing system, exemplified by Graylog, operates through a combination of various components, including Input, Extractors, Stream, and Index, with the latter being the most crucial. Input is a log data capture and acceptance component, while Extractors are log data format conversion components, allowing for log field conversions from different sources, such as converting Nginx status codes into English representations. Stream is a log information classification and grouping component, and Index is a data storage component. Converted data can be grouped into different Streams by different tag types, and these log data are stored in the specified Index database for persistent storage. In terms of simplicity, Graylog is similar to Snort and uses a modular approach, expanding its functionality with plugins. The framework is easy to understand and easy to expand. Graylog also provides a web interface with a web interface, which supports log data visualization and allows users to search and analyze log data, as well as customize rules. The flexible plugin system also enhances system availability. However, the disadvantage is that the interaction between the Graylog system and users needs to be completed in a command line form, which undoubtedly reduces the user-friendliness of the system.

OpenVAS is an open-source vulnerability scanning tool and a branch of the Nessus project. It is designed to identify security issues on a target network or host. Later on, the vulnerability scanner was developed and improved independently with a B/S architecture based on the earlier work. It performs scanning and provides the results of the scan. Its core component is a server that includes a set of network vulnerability detection programs that detect security issues in remote systems and applications. This server grants the user permission to run several different network vulnerability tests (written in the Nessus attack scripting language). In terms of simplicity, vulnerability scanners are generally simple in structure, with a regular and organized system structure, while being easy to extend with additional functionality. Furthermore, OpenVas has a graphical front-end, and nearly all operations can be done in the graphical interface, which is user-friendly. However, at the same time, due to the era, the graphical interface is rather old, which affects its usability. The more huge vulnerability database and system also increase the overall maintenance difficulty.

Authentication and access control are currently the most widely used methods for managing network security threats, and they are utilized extensively. The simple core features also make the system framework simple, and easy to expand and maintain. Due to their widespread use, the most commonly used authentication systems today have a simple and user-friendly graphical interface with excellent availability due to frequent updates and developments.

Honeypots are a relatively simple network security technique. The greatest advantage of honeypots compared to other security measures is their simplicity. They attract attackers

by mimicking one or more vulnerable hosts or services, capturing attack traffic and samples, discovering network threats, and extracting threat characteristics. Honeypots do not involve any special computing and do not require feature databases such as IPS and vulnerability scanners, or rule libraries such as Snort. All users need to do is place the honeypot in the system environment and run it, with excellent availability and scalability. The honeypot system is relatively easy to maintain and does not face the various threats that more complex security tools do, such as incorrect configuration, system crashes, and failures.

In conclusion, both identity authentication and access control systems, as well as honeypot systems, offer a significant advantage in terms of simplicity. However, intrusion detection systems, intrusion prevention systems, log auditing systems, and vulnerability scanners pose a significant disadvantage in terms of system simplicity due to the complexity of their installation and use processes.

*2.5. Summary*

Through the research and comparison of different network threat management methods, we find that different network threat management methods have different advantages in different aspects. For example, honeypot systems have great advantages in terms of attractiveness and simplicity, but have certain security defects; while intrusion detection systems have great advantages in terms of security and simplicity, but have serious shortcomings in terms of attractiveness. Therefore, the emergence of a network threat management system that takes into account various aspects such as interactivity, attractiveness, security, and simplicity is particularly urgent and important.

## 3. Central Node Model in High-Interactivity Honeypot System

In this paper, the high-interactivity honeypot system borrows the structure of the kernel-based operating system and adopts a modular design to complete the central node part of the high-interactivity honeypot system. The modular design is advantageous in several ways. Firstly, it enables team division of labor and cooperation, allowing team members to clearly understand their own work content. Secondly, it is more conducive to system debugging during the development stage, allowing targeted modification of a part of the system without modifying the entire system. Finally, the modular design is beneficial for adding or deleting functions based on the original system. The system has a central node serving as the connection node and information processing center for various high-interactivity honeypots. Based on the central node, the main components of the high-interactivity honeypot system framework are divided into information capture, connection control, honeypot deployment, data analysis and processing, and data storage. The operational structure of each part of the framework is shown in Figure 1.

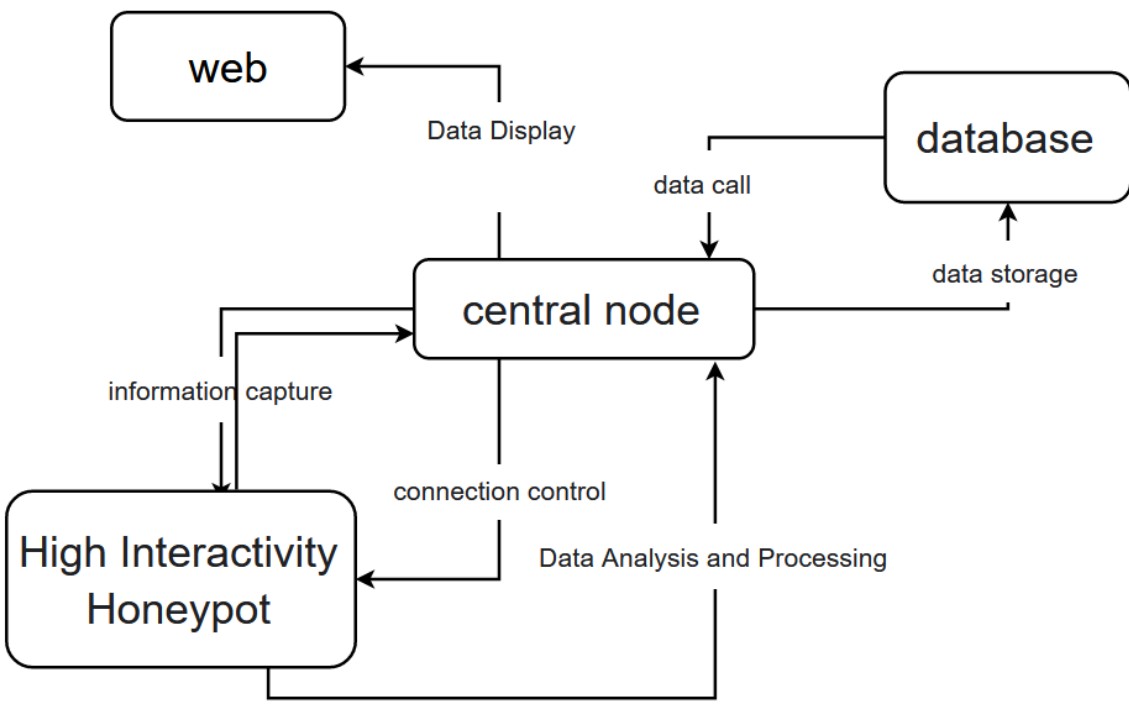

**Figure 1.** High-interaction honeypot system framework.

*3.1. Information Capture Module*

As one of the most core components of the honeypot system, the performance of the information capture module is a key indicator to determine the effectiveness of the whole honeypot system.

In our system, we use high-interactive honeypots for information capture. High-interactive honeypots provide a realistic operating system environment and can provide more attack information than low-interactive honeypots, and are less likely to be detected by attackers. After being subjected to a network attack by an adversary, a high-interaction honeypot acquires and records the attack data, compiles it into a package, and submits it to the central node for analysis. We use the form of loadable kernel modules (LKMs) to run internally in the system to avoid being easily discovered by the attacker and causing exposure to the honeypot system. If the honeypot is compromised by the attacker, the user can also disconnect the interaction between the central node and the honeypot to ensure the overall security of the system.

As shown in Figure 2, the high-interactivity honeypot generates an attack environment for simulated attacks. After the attacker launches an attack on the attack environment, the central node in the honeypot generates and packages the attack information. Once the packaging is complete, the specific attack information is transmitted to the central node of the entire system for further information processing.

It is noteworthy that to maintain a high level of interactivity within the honeypot system, a realistic operating system environment is provided to the honeypot kernel. Additionally, the honeypot kernel furnishes a dependable operating system for the simulated attack environment generated by the honeypot. In this way, the attacker is not easily aware that they are attacking a honeypot when carrying out an attack. Although this poses a certain challenge to the security of the entire system, it also protects the confidentiality of the entire system and the authenticity of the captured attack information.

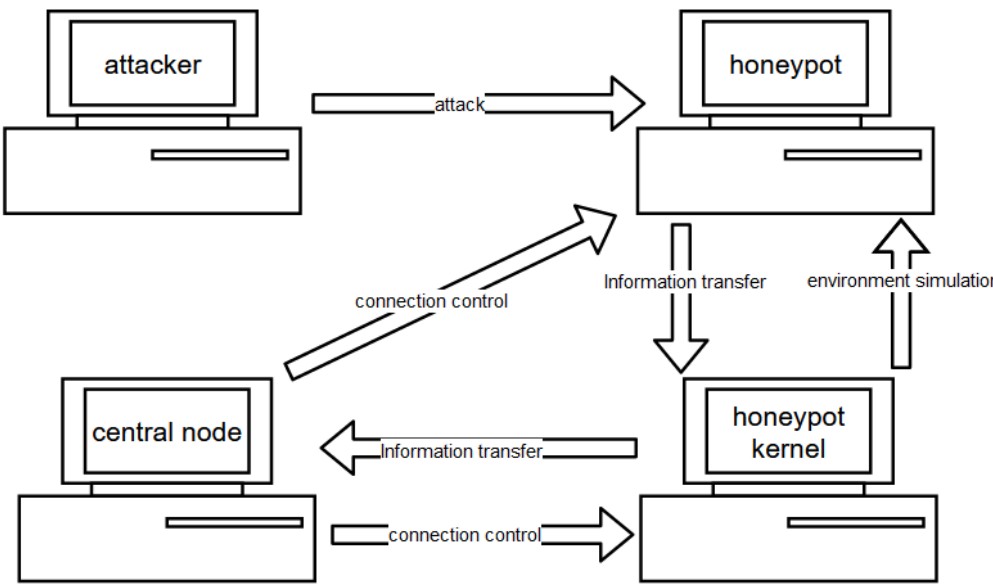

**Figure 2.** Information Capture Module It is worth noting that in order to ensure high interactivity.

*3.2. Connection Control Module*

To ensure high interactivity and strong control capability of the central node for the overall system, we design the connection control module as a central kernel-style structure that includes a communication control module and an environment control module.

3.2.1. The Communication Control Module

The communication control module mainly includes external connection restrictions and data packet suppression functions. We have enhanced the central node's control capabilities over the numerous highly interactive honeypots linked to the system. This upgrade allows us to monitor real-time status of each honeypot and guarantee a realistic simulation environment for them. More importantly, it ensures the overall security of the system.

External Connection Restriction: We have imposed restrictions on the honeypots that the central node must connect to, in order to enhance the security of the central node. We use login identity authentication control to ensure that the user of the honeynet is a "secure user" who is securely authenticated. IP address filtering is utilized to ensure that only legitimate honeypots are connected to the system. This filtering mechanism also helps in filtering out the collection of redundant information and prevents untrusted sites from accessing the honeypot system. To ensure that the log file recording hacker behavior is not illegally tampered with by hackers, we provide the ability to change the default storage path of the IIS log. These measures guarantee the security of the honeypot system itself. Furthermore, because the stored files themselves are "baited", when the hacker wreaks havoc on the website, he may have unknowingly fallen into being countered.

Data Packet Suppression Function: We filter and suppress the data packet information from the honeypots to the central node, in order to filter out useless information and potential attacks on the central node. The central node receives and stores data packets, which are initially "baited" packets that have undergone specialized processing by the honeypots [11]. The central node then proceeds to authenticate the IP and verify that the packet format meets its requirements before accepting the data for further processing. The specific workflow is shown in Figure 3 below.

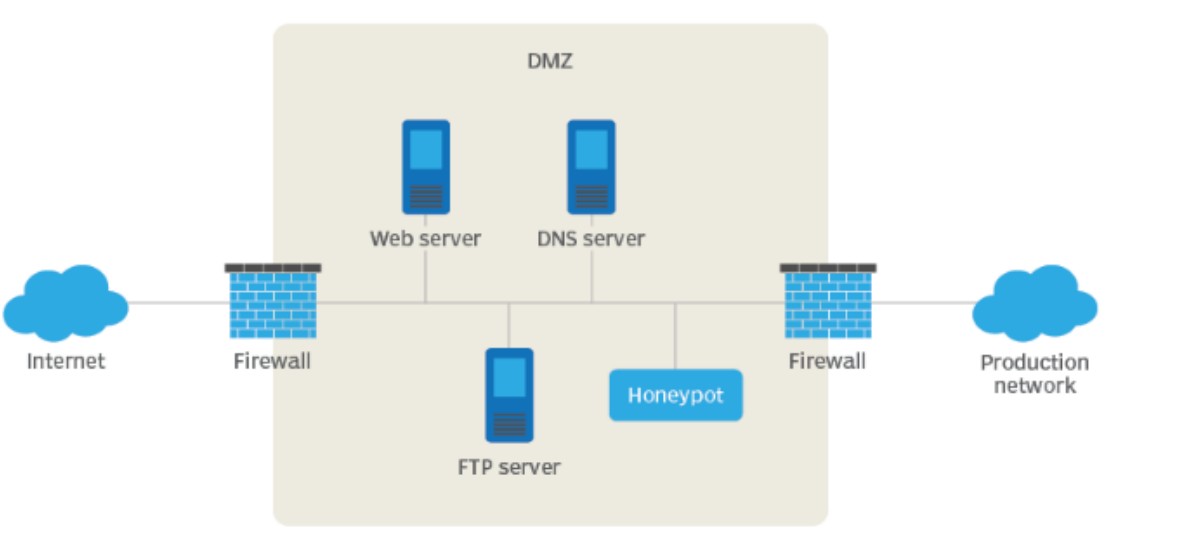

**Figure 3.** Honeypot in Network Architecture for Data Connection Control.

### 3.2.2. Environment Control Module

The Environmental Control Module aims to provide a realistic and reliable operating system for honeypots, ensuring high interactivity of the honeypot to obtain more attack information compared to low-interactivity honeypots, and improving the stealthiness of the honeypot system against attackers. The system does not merely imitate protocols or services; rather, it functions as an actual attack system, which substantially decreases the probability of attackers detecting that they are being redirected or monitored. Since the system solely serves as bait, any traffic detected is inherently malicious, thereby facilitating the identification of threats and enabling the monitoring of attackers' activities. This module can give insight into the tools that attackers use to escalate privileges, or their lateral movements in an attempt to discover sensitive data. We adopt state-of-the-art dynamic deception methods to ensure that the highly interactive honeypots can adapt to each event and keep attackers unaware they are using bait. The specific workflow is shown in Figure 4 below.

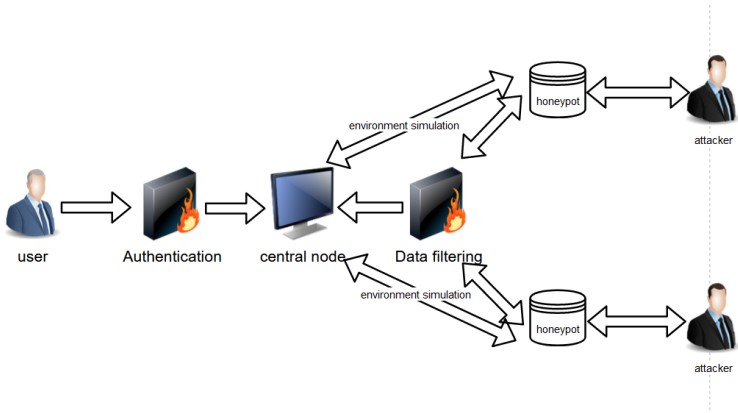

**Figure 4.** The overall structure of the connection control module.

### 3.3. Honeypot Deployment

The high-interactivity honeypots we use in the system are usually based on real systems and are physical honeypots rather than simulated honeypots. The attackers are confronted with genuine systems and services that can amass greater amounts of information and exhibit more advanced levels of interaction and interactivity with the

attackers. The attackers can detect, attack, and destroy these systems and use them as tools for further attacks. Compared to low-interactivity honeypots, high-interactivity honeypots are more complex in configuration, require more resource support, and have higher risks, but relatively, they can also collect more valuable information and have a higher degree of concealment for the attackers. The specific workflow is shown in Figure 5 below.

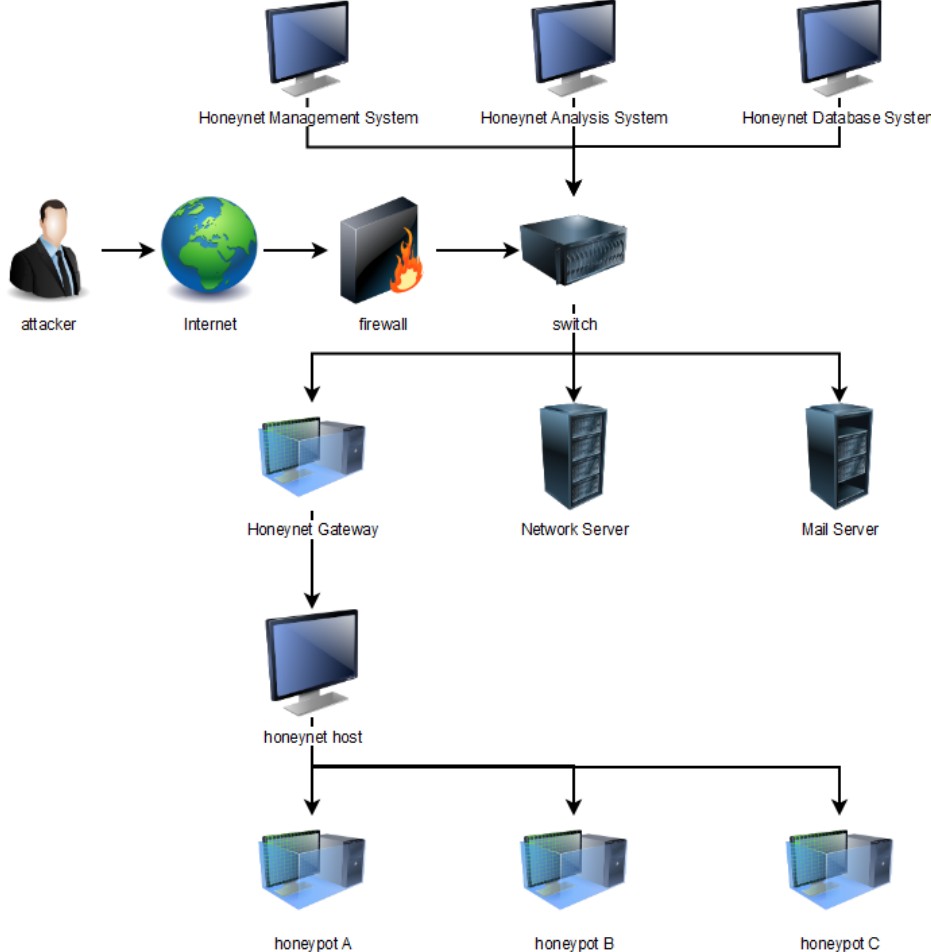

**Figure 5.** Honeypot deployment module model.

The honeypots connected to the system are directly provided by the central node with a simulated operating system to ensure high interactivity. Prior to deployment into the system, the central node will conduct security checks on the honeypots that seek to join, ensuring their security and compliance with the central node's requirements. This process aims to prevent potential threats to the system's overall security and stability, as well as to maintain the honeypot network's effectiveness in detecting and mitigating attacks. After connecting to the honeypot system, the central node will assign a specific IP to the honeypot for internet access. As part of its operation, the system administration module deploys "baited" files on publicly accessible web and FTP sites, commonly known as "baited" sites. In the event that an external connection directly accesses these "baited" sites or attempts to probe the server group and gets redirected to a "baited" site, the system's "baited" files will trigger a response to counteract the attempted intrusion.

## 3.4. Data Analysis and Processing

The data analysis and processing module is one of the central node's most critical functions, as it plays a pivotal role in determining the overall performance of the high-interactivity honeypot system. Thus, it is a key area of focus in our system's design and

implementation. We divide the data analysis and processing module into attacker information formatting function, information classification function, and risk assessment function.

### 3.4.1. Attacker Information Formatting Function

To ensure uniform information processing caliber and simplify the operation of honeypots joining the system, we have standardized the attacker's attack information. On the honeypot side, the attack information, comprising both attacker and attack data, is standardized and subsequently transmitted to the backend for further processing.

### 3.4.2. Information Classification Function

In order to ensure the efficiency of information retrieval at the central node, we have centered the information retrieval function at the central node around the attacker's IP. If you want to query specific attack information, the central node will perform information retrieval and display based on the attacker's IP as the core. This information includes the attacker's attack operations, attack score, and other related attack information. The specific workflow is shown in Figure 6 below.

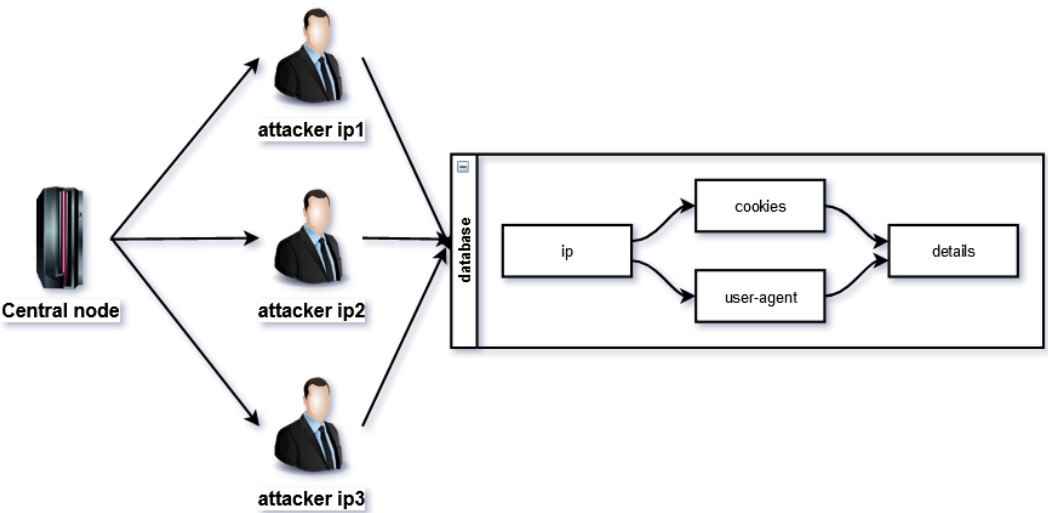

**Figure 6.** Information Classification Query Model.

### 3.4.3. Risk Assessment Function

In order to make the threat level of an attacker's attack more intuitive, the system has designed a risk assessment function that scores each threat attack by the attacker. The higher the score, the higher the threat level of this attack. See Section 4 of the paper for the specific implementation.

### 3.5. Data Storage

In order to make the back-end debugging work easier and the data display more intuitive, we use the MongoDB database for data storage [12]. We divide the database into four tables: user table (User), honeypot table (Pot), information table (Information), and attacker table (Attacker) to store information.

User table: including username, password, role <guest, admin> several items, mainly used to store the information of users who use the honeypot.

Honeypot table (Pot): including honeypot id (potId), honeypot ip (potIp), honeypot type (potType), honeypot private key (potkey), and status, which are stored and connected to the central node The specific information of several honeypots. Every time a honeypot sends information to the central node, it needs to verify the information in the honeypot table.

Information table (Information): including honeypot id (potId), time, attacker ip(attackerIp), attack type(attackerType), attack information (attackInformation), and attack

score (attackPoint). It is mainly used to store the specific information of each network attack on the system and the scoring and evaluation of the attack by the central node.

Attacker table (Attacker): including attacker ip (attackerIp), attacker information (attackerInformation), attack times (attackNum), attacker score (attackerPoint), and status. It is used to save the specific information of the attacker who launched the attack on the honeypot system.

## 4. Implementation and Results Analysis of High Interactivity Honeypot System

A honeypot is a specialized system designed to attract attackers by simulating a real system and intentionally containing a significant number of vulnerabilities. The primary objective of a honeypot is to entice attackers to engage with the system, allowing security professionals to observe and analyze their behavior in order to enhance security measures for the real system. The log records left by various security devices or attackers can be analyzed to restore the entire attack path of the attacker and the attack load used by the attacker, and the honeypot plays two important roles in this process. One is it is to induce the attacker to attack a false and non-production system, which consumes a lot of time and experience of the attacker and reduces the threat to the actual system. Another benefit of deploying a honeypot is that it can aid security personnel in tracing back to the attacker through intrusion analysis. By collecting attack evidence, security professionals can use the information to pursue legal action against the attacker. Additionally, by analyzing attack data, defenders can identify characteristics of the attack before it occurs and use this information to address vulnerabilities in advance. Honeypots can be roughly divided into three types according to their degree of interaction: low-interaction honeypots, medium-interaction honeypots, and high-interaction honeypots. Among them, low-interaction honeypots and medium-interaction honeypots only provide fake services to collect attacker information. A low-interaction honeypot is designed to limit the interaction between a hacker and the system, minimizing the opportunity for a black hat to invade the system. Due to its limited functionality, the attacker's interaction with the system is brief and restricted, reducing the risk of harm to the honeypot and the network it is designed to protect. This type of honeypot can greatly protect itself from intruders. However, such honeypots get little information about hackers. Therefore, this approach is widely used by companies that care about protecting their systems from the Internet [13]. The highly interactive honeypot provides the real system or operating environment and records the operating records of the system. In contrast, highly interactive honeypot systems are more stealthy and more difficult to deploy. In a high-interaction honeypot, the main focus is to obtain maximum information about the hacker, allowing them to access the entire system and even tamper with it [13]. The highly interactive honeypot network implemented by this system needs to achieve honeypot interaction, security patches, risk convergence, full logs, high forgery degree, self-protection, version selection, and other characteristics in common scenarios [14]. In order to make the highly interactive honeypot of this system meet the above requirements, the design starts from the overall structure of the system and then analyzes and designs the realization of the honeynet central node and the highly interactive honeypot, respectively. Finally, based on the system design, the current high-interaction honeypot system is tested and analyzed. The comparison is shown in Table 1 below.

**Table 1.** Comparison of interactive honeypots.

| Interaction Level | Installation Configuration | Deployment Maintain | Information Collected | Risk Level |
|---|---|---|---|---|
| Low | Simple | Simple | Limited | Low |
| Medium | Involved | Involved | Variables | Medium |
| High | Difficult | Difficult | Rich | High |

*4.1. Implementation of Honeypot Central Node*

For the design and implementation of the central node, it must ensure its high level of fakery and self-protection. Although maintaining a high level of fakery is more of a characteristic of a honeypot, the central node also relies on a high level of fakery to ensure that it is not accidentally recognized and attacked by attackers. Due to the overall design architecture, once the central node is attacked, the entire honeypot system will be impacted. In order to achieve a high level of authenticity for the central node in a honeypot system, all external network connections must be strictly restricted. Only the communication interface and WEB management interface of the honeypot should be exposed, and there should be no other external connections. This approach will help to ensure that the central node appears as realistic as possible to potential attackers. Secondly, in order to prevent attackers from infiltrating the central node through the honeypot, the central node needs to deploy a protective system consisting of a firewall and intrusion detection system to achieve self-protection. The specific workflow is shown in Figure 7 below.

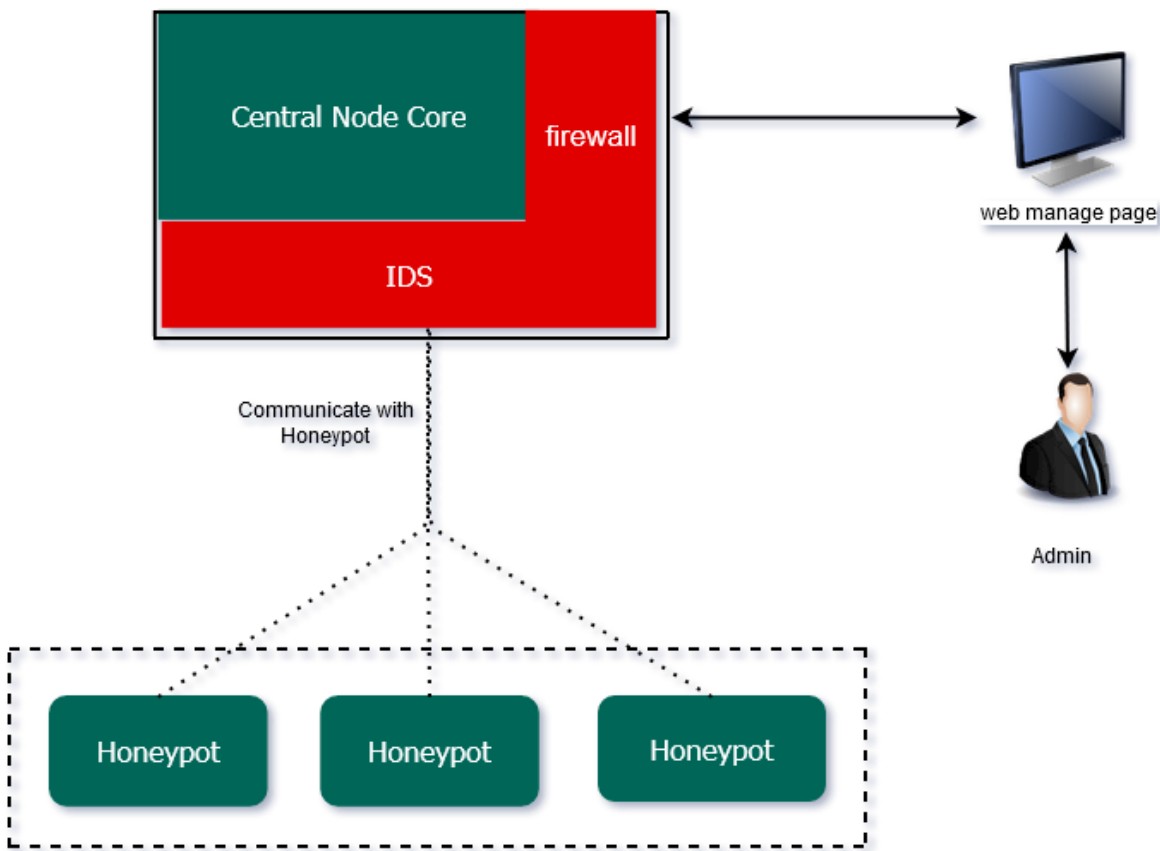

**Figure 7.** Central Node Interaction Model.

In our implementation, strict restrictions are imposed on the content security policy at the backend to ensure secure interaction between the front end and back end. Only API access from a trustworthy front end is processed. All the honeypots are stored as tuple data in the central node, consisting of the honeypot IP address, a randomly generated unique key assigned to each honeypot, and other relevant information. This random key satisfies the characteristics of a universally unique identifier, ensuring uniqueness and reliability. This key is kept secret in the central node and there is no way for it to be disclosed after it is created. This key serves as the unique master key for communication between the central node and the corresponding honeypot. All communication traffic is encrypted using the session key generated by this master key in CBC mode AES encryption. To

facilitate critical interactions between the central node and honeypots, a distinct backend API is utilized. This API accepts encrypted data along with the honeypot IP address. The central node then maps the IP address to the corresponding universally unique identifier, decrypts the encrypted data with the matching session key utilizing the CBC mode AES decryption algorithm, and verifies the data's validity only if it conforms to specific JSON object properties. Any other format data or decryption failure is considered an abnormal interaction and triggers the corresponding exception handling module, rolling back any information saved in the process. The specific workflow is shown in Figure 8 below.

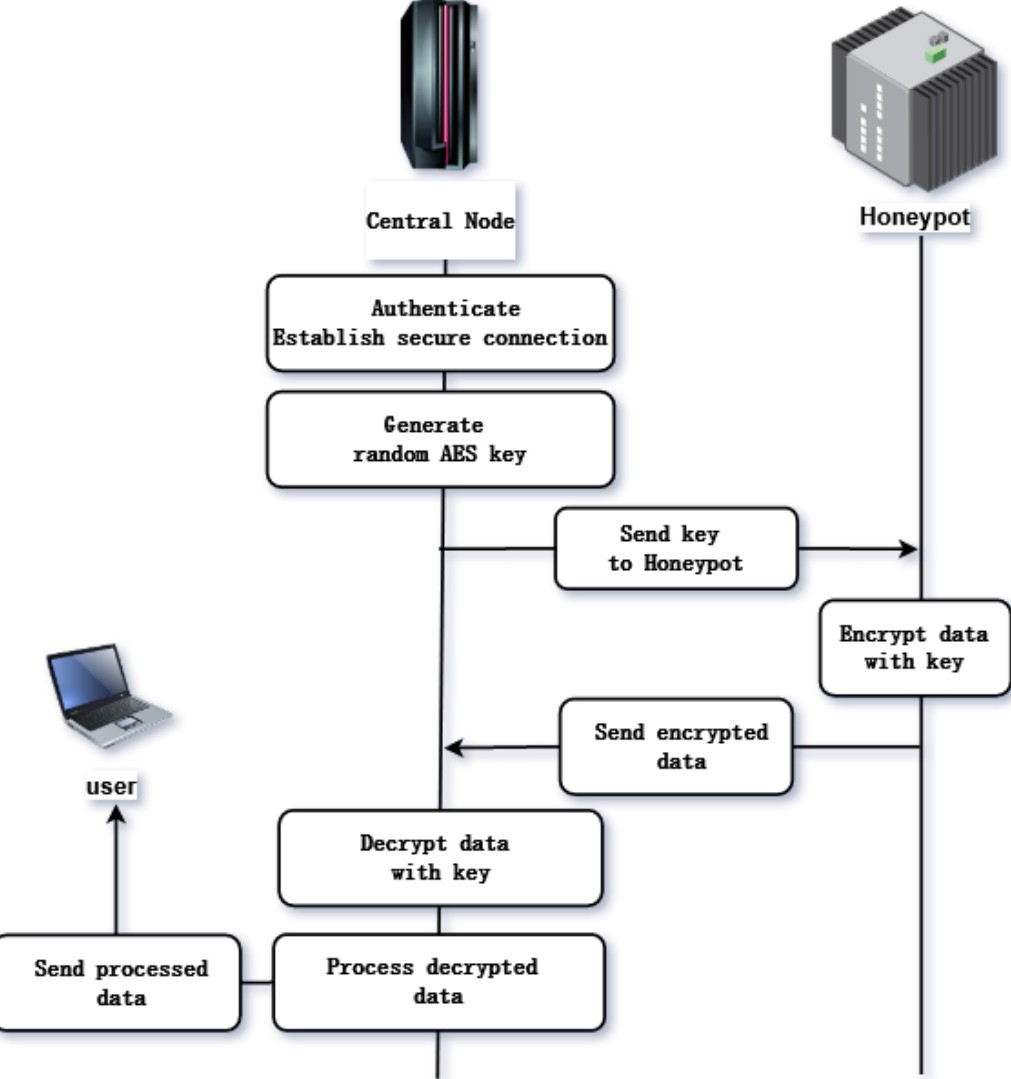

**Figure 8.** Central node data flow model.

After receiving a legitimate honeypot interaction, the central node invokes the core data cross-processing computation method, which is detailed in this paper. The newly obtained information is analyzed from multiple angles and stored in the database. The core data cross-processing method involves passing in the authenticated JSON data returned by each honeypot, which contains key-value pairs. A loop is used to iterate through the keys, and if a key corresponds to a value that is a JSON object, it is recursively iterated through. Once all the values are available, they are aggregated into a key-set. This is the pre-processing step for a copy of the honeypot incoming data. All incoming data must first be preprocessed as JSON objects. In this way, we obtain a collection consisting of a key table of keys of individual JSON objects. By performing different set operations on the elements

in the set, the key table can be queried from a specific perspective, and the results obtained will have specific features. The results are the mapping of the whole dataset under this feature, which is saved in the database and bound to the feature value for the next query. The next time a new dataset is imported, we only need to preprocess the new dataset and perform the set operation on the new key table set again to update the results corresponding to the features. The newly acquired information is subjected to multidimensional analysis and is cross-stored. Subsequently, a multi-dimensional attack topology map is generated with the attacker and honeypot at its center. The map provides insights into the attacker's attack path and tactics. Furthermore, it evaluates all the recorded attacks on the honeypot.

*4.2. High Interaction Honeypot Implementation*

For the implementation of highly interactive honeypots, it is necessary to implement a full amount of logs, a high degree of forgery, self-protection [15], version selection, and other features. The full amount of logs is the cornerstone of the honeypot system. Adequate collection and preservation of system operation logs, alarm messages, and error messages are necessary to identify potential threats to the honeypot and collect sufficient information for analysis to be sent back to the central node. In our honeypot, we choose to record comprehensive and complete log information to ensure a thorough analysis. To achieve this, we record program runtime logs in both the high-privilege kernel state and low-privilege user state. This includes, but is not limited to, runtime messages, warning messages, error messages, output streams, input streams, and error streams. High interactivity is guaranteed by high forgery, and if an attacker suspects that the system is a honeypot, then the collected information becomes meaningless. In our honeypot, we run a complete and genuine system, which is fundamentally not forged and thus naturally has high interactivity. Self-protection is to protect the security of the whole honeynet. Only when the honeypot has a sufficiently high level of security line controllability can it prevent the attacker from breaching the central node and other honeypots through the honeypot itself. To achieve this, we utilize a daemon within the honeypot to execute the service and lock the process in the event of abnormal termination, preventing its automatic restart and any network interactions. Version selection is then to allow easy and fast construction of patterned honeypots. Modern products are characterized by frequent version updates and lower security line stability, making them more vulnerable to attacks by threat actors. In the case of highly interactive honeypots, creating new versions of the honeypot system that align with the attacker's preferences is necessary, even if the honeypot lacks support for version selection, which can be challenging to achieve. Virtualization technologies such as Docker can be utilized to rapidly deploy a range of honeypots with distinct versions but identical functional structures via configuration files, thereby supporting version selection. The specific workflow is shown in Figure 9 below.

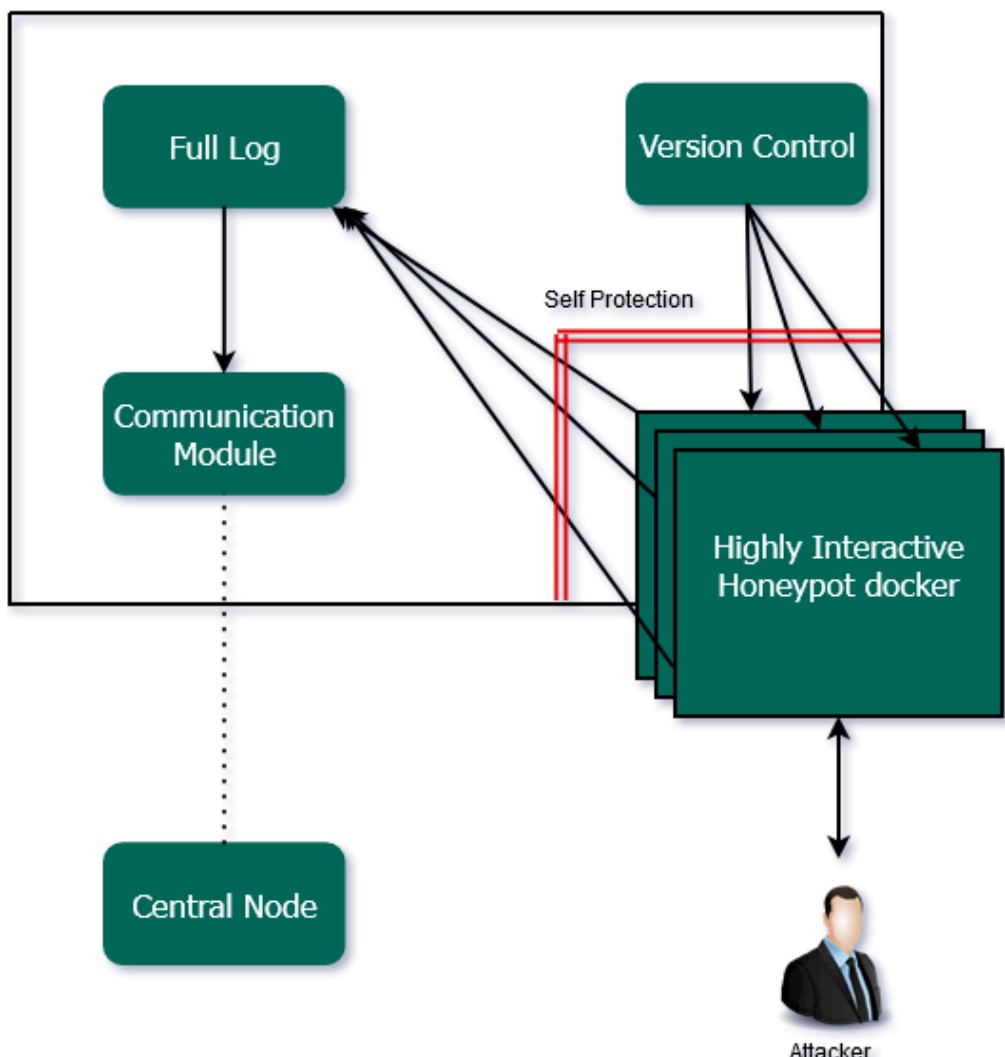

**Figure 9.** Schematic diagram of attack.

In terms of specific honeypot selection, we used Shadow Daemon and Ehoney as our concrete implementation targets.

Shadow Daemon is a collection of tools for detecting, recording and preventing attacks on network applications. Technically, Shadow Daemon is a network application firewall that intercepts requests and filters out malicious parameters. It is a modular system that separates web applications, analysis, and interface to improve security, flexibility, and scalability. The software can accurately identify attacks such as SQL injection, XML injection, code injection, command injection, backdoor access, etc., and accurately record them. This firewall also incorporates a honeypot. Although most honeypots on the market excel in attracting and documenting attacks, dynamic network applications expose significant information compared to conventional network services. As a result, it is easy for hackers to determine if the application is actively running or if the website's behavior appears unnatural, thereby revealing that it is a honeypot and deterring attackers from taking action. Thus, the existing solutions are primarily useful for gathering information on trusted self-propagating malware, which represents only a fraction of all attacks. Consequently, the collected data is incomplete and non-conclusive. To overcome this challenge, the system must be capable of detecting and logging malicious requests on the production server, as the authentic target is the only reliable source of information. The collected information is more meaningful and suitable for research because they are not distorted.

This honeypot has many advantages. Firstly, by default, Shadow Daemon acts as a web application firewall and blocks malicious requests, but it is also designed to be used as a high-interactive honeypot. With a few simple modifications to the configuration file, the protection of the honeypot combined with the firewall can be disabled, allowing the system to remain hidden in the shadows. This modification increases the efficiency of the firewall and makes it harder for attackers to detect it as a honeypot, thus making it easier for the user to collect attack records. Additionally, the combination of the honeypot with the firewall is easy to operate and generates fewer errors in the collected data, making the data more authentic. It can be combined with other honeypot systems and integrated into a honeynet as an additional source of information [2].

However, the honeypot still has many disadvantages: first, the software itself is a firewall, and protection functions of the firewall need to be disabled for the honeypot to work normally. If the system does not have another firewall to protect it, the system is likely to crash and fail to achieve the purpose of the honeypot. Secondly, as a single honeypot, it lacks attraction to network attacks and attackers lack the motivation to attack. It needs to be combined with other honeypots to form a honeynet to effectively play its role. Finally, the configuration process of the honeypot is complex and there is a certain threshold for new users to use it.

In response to the shortcomings of this honeypot, we have made corresponding improvements: we have separated the system firewall from the honeypot, so the honeypot does not need to close the firewall during operation and have securely authenticated the high-interaction honeypot through the central node, which issues authentication and registration information to the honeypot to ensure that the system is adequately protected while collecting data. Furthermore, we have streamlined the configuration process of the honeypot by implementing a more lucid and user-friendly client interface. This modification has resulted in a lowered barrier to entry for new users. Additionally, we have standardized the communication protocol between the honeypot and the central node, using a unified packet format that allows for effortless customization based on individual requirements. Finally, in response to the issue of weak attraction of this honeypot, we have strengthened its attraction as an individual honeypot by designing more enticing vulnerabilities, such as SQL injection vulnerabilities, and expanded its information acquisition scope.

At the same time, we also adopted Ehoney as part of the system connecting to honeypots. Ehoney is the first open-source deception defense system from Seccome Teamer. It categorizes attacks into various phases, including reconnaissance, weapon development, payload delivery, vulnerability exploitation, implant deployment, ongoing control, and goal achievement. Moreover, the document outlines corresponding countermeasures to address each stage of the attack.

The advantages of Ehoney are as follows: Firstly, it is business-oriented, has a simple and user-friendly operation method, and has a clear front-end, allowing for easy deployment and viewing of honeypot attack information with just simple operations. Secondly, Ehoney has comprehensive and detailed documentation, making improvement, communication and sharing easier, increasing the program's scalability. Furthermore, this honeypot can generate various secret signatures, meaning that the characteristics of some accessible network resources from certain types of files can be modified to generate a secret signature. When a hacker downloads a document or accesses a folder from an infiltrated server or honeypot, an alarm is triggered. Finally, Ehoney utilizes topology visualization technology to present attack views visually, rendering all attacks perceptible, and constructing a comprehensive attack chain. This technology renders formerly opaque network defense and offense transparent, thereby facilitating analysis of the attacker's methods and objectives.

The disadvantages of Ehoney are also quite obvious: First, Ehoney is a high-interactivity honeypot system, but due to its high interactivity, it may be exploited by hackers to carry out lateral exploration and gain root privileges to achieve their ultimate goals. Secondly, Ehoney lacks honey bait, making it weaker in terms of luring attackers. Finally, while a

single Ehoney honeypot may not be highly tempting, deploying too many can increase the burden on the server.

In response to the disadvantages of Ehoney, we have made corresponding technical improvements: first, while improving the interactivity of the honeypot, we have strengthened the information isolation between the system and the honeypot to prevent the honeypot from being used by hackers in reverse. Secondly, we have incorporated additional highly enticing vulnerabilities as honeypot bait, ensuring that in the event of a honeypot breach, attackers will not promptly exit and initiate a real server attack, but instead necessitate time to analyze the acquired data.

*4.3. Overall Structure Design Implementation*

Among the features of a honeypot system listed above, risk convergence, self-protection, and version selection mainly rely on the overall system architecture design. Risk convergence refers to the fact that the entire system should only expose pre-set vulnerable ports or services while minimizing the possibility of exposing irrelevant services to the internet surface. This ensures that the attacker's attack surface only includes the honeypot's pre-set vulnerabilities, thus avoiding the possibility of attackers using other methods to unintentionally breach the honeypot. For the overall structure of a highly interactive honeypot network with the central node as the center and each honeypot as the endpoint, the expected vulnerability interface of each honeypot should be the only external exit, and other non-vulnerability-related interfaces, such as those interacting with the central node, should not be exposed to the Internet surface for an attacker to access.

In response to the above problems, we adopt a modular approach to the system as a whole, so as to facilitate the early development of the system and the later function deletion. We design the core of the entire system as a modular honeynet central node, which serves as the exchange, storage, and processing center of the overall data flow of the system, as shown in Figure 1. The specific implementation can be found in the first part of this section. We design the information capture module of the system as a high-interaction honeypot connected to the central node. To enhance concealment and collect more detailed attack information, we provide real services. Please refer to the second part of this section for specific implementation details. The information display component of our system is designed as a front-end display page, which shows the attack information saved by the attacker after attacking each high-interaction honeypot. It also displays the analysis of the attack information generated by the back-end, assisting users in identifying attack characteristics, fixing system vulnerabilities, and providing a detailed discussion of the information display component's implementation.

Considering the protection of users' personal information, the project team first designed the login and registration interface of the system to ensure the safety of users' personal privacy. After being authenticated by the system, users can use the product. We have divided the information to be displayed into three parts: information display, status adjustment, and information statistics. The information display interface is shown in the picture. In the attack information part, the ip, attack times, attack score, and detailed information are displayed in detail; in the honeypot information part, the honeypot ID, IP, Honeypot type, and other information. Among them, the honeypot ID is automatically generated and issued by the central node after authentication, and the attack score is scored by the central node after assessing the threat of the attack to help users assess the threat of the attack. The specific workflow is shown in Figure 10 below.

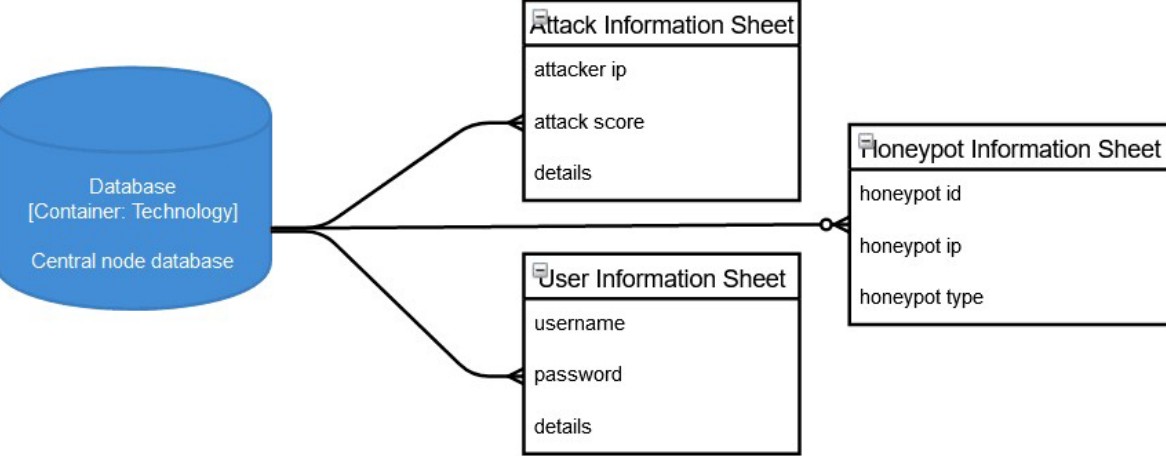

**Figure 10.** Database structure.

In the attack information section, detailed information such as honeypot status and attacker status can be hidden and removed from the honeypot connected to the central node. Hidden means temporarily disconnecting the honeypot from the service, but the honeypot is still connected to the central node and can be online at any time; removed means disconnecting the central node from the honeypot and disconnecting the honeypot from the service, and if the service needs to be online again, it needs to be connected to the central node again. The specific workflow is shown in Figure 11 below.

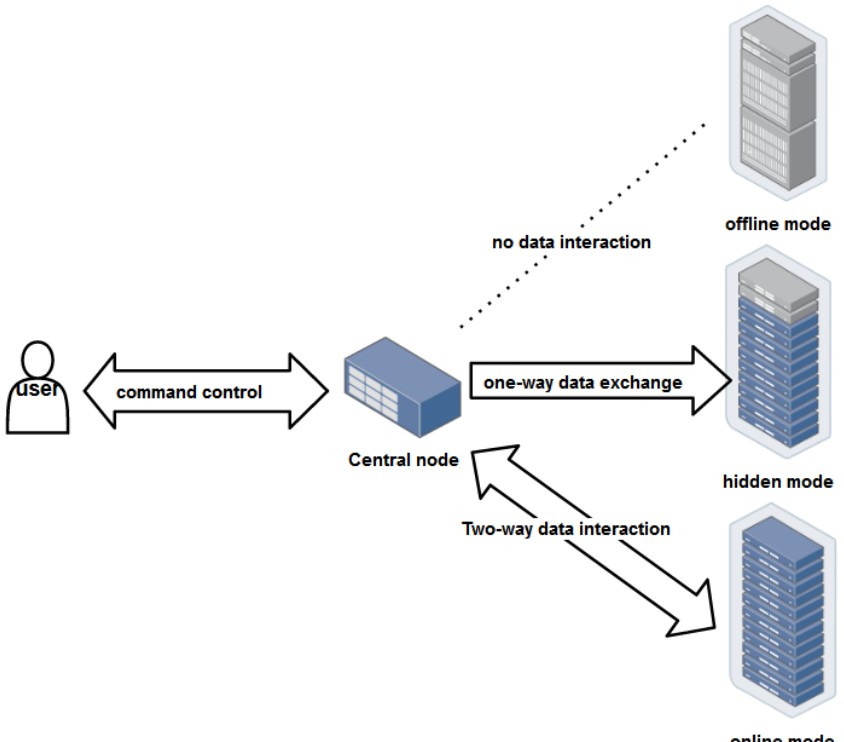

**Figure 11.** Honeypot state control.

*4.4. Honeynet Test Analysis*

With its features including security patch, risk convergence, comprehensive logging, high forgery level, self-protection, and version selection, the honeypot has successfully achieved its fundamental functions while maintaining a high level of disguise and interactivity. Through the testing of the central node and the modules of the honeypot, it can

be found that the modules work well with each other, and the attacker can only leave the information of his own attack through the docker service of the highly interactive honeypot, and the attack information is finally collected through the full amount of logs and sent to the central node after encryption, and the central node can display the summary on the website management page after processing the information. The central node itself improves its own security through an intrusion detection system and a firewall.

We put the system online in the public network, assigning ports to different highly interactive honeypots to run and attracting attacks from the public network, while project team members also attacked the honeypot system to test its security, leaving the specific attack information as shown below, the attacker's IP address, the number of times attacked, the attack score and the specific information left by the attack can be displayed completely and intuitively. The results are shown in Figure 12.

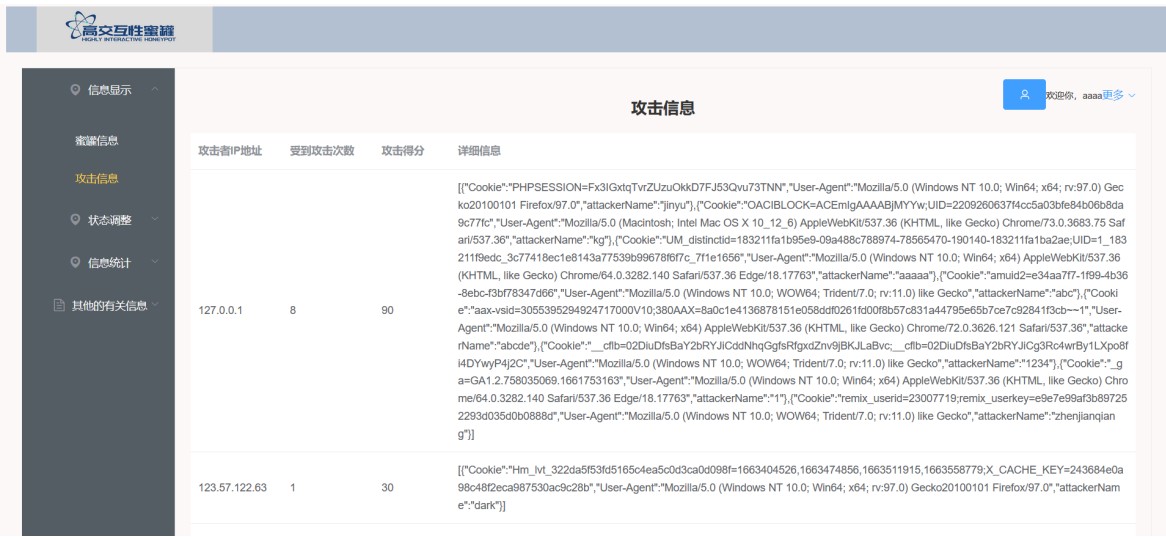

**Figure 12.** Details of attack information.

## 5. Data Comparison with Various Cyber Threat Management Methods

Through the operation and maintenance of the public network for a period of time, the network threat management platform in this paper has collected a total of 65 attack data, the results are shown in Figure 13.

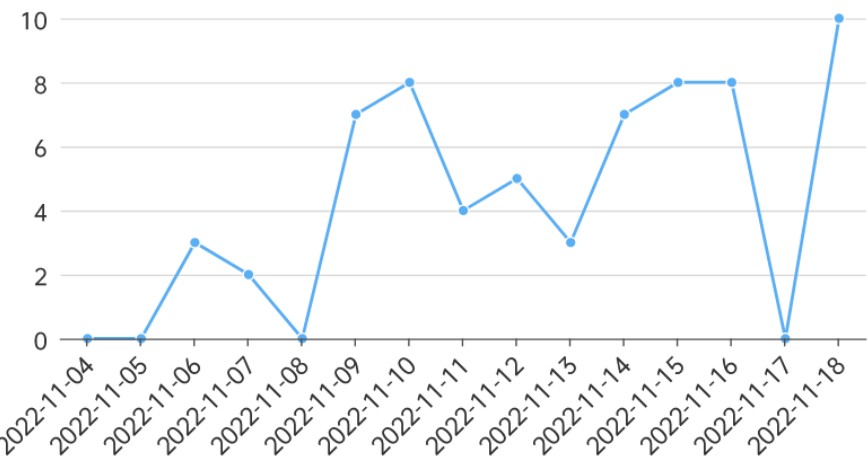

**Figure 13.** Attack Frequency Chart.

The network threat management system presented in this paper is subject to continuous attacks from the network platform on the public network. However, the central node remains uncompromised throughout the attack process, which is a clear demonstration of the effectiveness of the network threat management tool developed using this methodology. This achievement underscores the system's ability to provide security and resilience in the face of malicious attacks, thus elevating it to a higher level within the current network environment.

In this highly interactive honeypot system, the honeypot can accurately collect the IP address of the attacker, the number of times the attacker uses the IP address to attack the honeypot, and the score of the attack, when an attacker attacks. These details can help in determining the threat posed by the attacker and the target. Furthermore, the attacker's springboard machine, meat machine, and even the attacker himself can be tracked through the IP address. Additionally, the honeypot provides some details about the attacker, such as the cookie used by the attacker, the operating system information used by the attacker, the attacker's browser information, and the time of the attack. By analyzing these details, users can develop insights about the identity and behavior model of the attacker and provide some help for attack countermeasures. The information collected is shown in Table 2.

**Table 2.** Data Information Collection.

| Types of Information Collected | How Information Is Used |
|---|---|
| IP address | Query the attack information by consulting the threat intelligence agency<br>Through the attack payload, you can obtain the information of the attacker's springboard machine<br>Collect information on attack IP assets through cyberspace search engines<br>Obtain attacker information by counterattacking IP addresses<br>Obtain the attacker's social information or countermeasure through the IP address<br>Obtain the attacker's information through the domain name |
| Number of attacks | Determine the attacker's desire to attack<br>Determine the direction of the attacker's attack |
| Attack score | Determine the threat strength of the attacker |
| Cookie | It provides a reference for determining the identity of the attacker |
| Operating system information | It provides a reference for determining the identity of the attacker<br>Countermeasures against the operating system used by the attacker |
| Attacker browser information | It provides a reference for determining the identity of the attacker<br>Carry out certain countermeasures against the browser used by the attacker |
| Time of attack | Enrich the life logic of the attacker according to the attack time |

At the same time, we will use a highly interactive honeypot network threat management system and the mainstream use of traditional network threat management methods on the market to compare various aspects (here the commonly used snort, suricata, bro, and this paper uses a highly interactive honeypot network threat management system for comparison). The specific comparison is shown in Table 3.

**Table 3.** Comparison Table.

| Parameters | Snort [2] | Suricata [2] | Bro [2] | A Network Threat Management System for Highly Interactive Honeypots |
|---|---|---|---|---|
| Supported Platform | Win, MacOS, Unix | Win, MacOS, Unix | Unix like system, MacOS | Win, MacOS, Unix like system |
| License | GNU GPL V2 | GNU GPL V2 | BSD | NULL |
| IPS feature | Yes | Yes | No | Yes |
| PGP signed | Yes | Not Applicable | No | Yes |
| Support to high speed network | Medium | High | High | High |
| Configuration GUI | Yes | Yes | No | Yes |
| Offline Analysis | Yes for multiple files | Yes for single file | Yes for single file | Simultaneously to many objects |
| Threads | Single Thread | Multithreaded | Single Thread | Multithreaded |
| IPV6 | Yes | Yes | No | Yes |
| Installation and Deployment | Easy | Easy | Difficult | Easy |
| Detection mode | Passive | Passive | Passive | Initiative |

As shown in the table above, the network threat management system (NTMS) proposed in this paper offers significant advantages over traditional NTMS. Specifically, the proposed NTMS boasts strong system applicability, accommodating a variety of operating systems including Win, MacOS, Linux, and Unix-like systems. Second, it closely fits the current state-of-the-art network environment. For the current Internet environment with large data volumes and complex network situations, this paper designs the system to be applicable to the IPv6 network, while multi-threaded detection improves the data processing volume. Furthermore, in order to facilitate users, this paper designs the system as a simple operation platform with much lower operation and installation difficulty than Bro and designs a graphical user interface, which is more straightforward and user-friendly compared to snort and suricata interfaces. Last but not least, the biggest advantage over traditional NTMS is that the use of honeypots turns attacks into passive ones, and collects attack data in advance by setting decoys, so as to reduce irreparable damage caused by hacker attacks early.

Compared with the current market mainstream use of medium and low-interaction honeypot systems, the high-interaction honeypot used in this paper has incomparable advantages, here we use the medium-interaction honeypot kippo [16], low-interaction honeypot baapp [17], and our high-interaction honeypot system for comparison, the specific comparison is shown in Table 4.

Through the actual test comparison of the three honeypots, we believe that the NTMS introduced in this paper has the following advantages: first, compared with the ordinary low- and medium-interaction honeypots, the system adopts a graphical interface, which is user-friendly and more guided, and greatly reduces the difficulty of installation and use; secondly, the high-interaction honeypot utilizes a genuine operating environment that is more enticing and perplexing for attackers. Consequently, when compared to low- and medium-interactive honeypots, the highly interactive honeypot system can gather more data in the same network environment. Additionally, because attackers have more comprehensive access rights, the highly interactive honeypot system can detect deeper attacks, obtain more complex types of data, and capture a wider breadth of data. This breadth of captured data is more conducive to enabling users to improve their systems based on the attack situation. Finally, in actual use, none of the three honeypots have been hacked, so the highly interactive honeypot system in this paper ensures the security of the system by connecting the central node to the real environment. In summary, this paper concludes that this network threat management approach is a great improvement over

traditional network threat management methods in terms of innovation, security, efficiency, simplicity, and accuracy.

**Table 4.** Comparison Table.

| Parameters | Baapp [17] | Kippo [16] | A Network Threat Management System for Highly Interactive Honeypots |
|---|---|---|---|
| Supported Platform | Linux | Win, Unix like system | Win, MacOS, Unix like system |
| License | V2 | V2 | NULL |
| IPS feature | Yes | Yes | Yes |
| PGP signed | No | Yes | Yes |
| Support to high speed network | Medium | Medium | High |
| Configuration GUI | No | No | Yes |
| Offline Analysis | Simultaneously to many objects | Simultaneously to many objects | Simultaneously to many objects |
| Threads | Multithreaded | Multithreaded | Multithreaded |
| IPV6 | No | No | Yes |
| Installation and Deployment | Difficult | Difficult | Easy |
| Detection mode | Initiative | Initiative | Initiative |
| Amount of data collected(7 days) | 23 | 18 | 65 |
| Detectable Attack Types | 4 | 4 | 8 |
| Attack source detection | No | Yes | Yes |
| Whether the attack depth can be detected | No | No | Yes |
| Attacker Behavior Detection | No | No | Yes |
| Data capture type | 4 | 4 | 8 |
| Attack duration | Unable to count | Unable to count | 173 min |

## 6. Discussion

After reviewing the current research and application status of honeypot technology, this paper proposes a modular design approach for developing a highly interactive honeypot threat management system. The proposed system model is achieved through the division of honeypot functions. This paper presents a detailed description of the core module (central node) of the system, encompassing the information capture module, connection control module, deployment of honeypots, data analysis and processing, as well as data storage. Furthermore, by building the model, the practical degree of the model is tested practically. It is considered that it basically achieves the expected goal of simplifying the operational difficulty of honeypot technology based on reducing the cost of using it so that it can be more easily used by SMEs and individual users. Of course, the research and implementation process also found shortcomings: when deploying multiple honeypots on the system, a specific analysis of multiple honeypots is required, so that a certain degree of encapsulation can be carried out to reduce the user's difficulty in using them. At the same time, there are various trends in the development of honeypot technology, such as honeypot-based security technology for even communication networks, honeypot-based DDoS attack defense technology, honeypot-based anti-phishing technology and honeypot-based big data technology, etc., which are also potential directions for our subsequent research. By deploying a range of novel honeypots, we anticipate that we can offer users a more comprehensive and secure protection system. This advancement will undoubtedly facilitate the ongoing evolution of honeypots and their associated technologies.

## 7. Conclusions

By analyzing the current state of honeypot technology research and application, this thesis adopts a modular design approach to create a highly interactive honeypot threat

management system model by breaking down the honeypot functions. This paper provides an in-depth account of the system's overall structure and the specific implementation of the honeypot, dividing the model into information collection, connection control, honeypot deployment, and data analysis and processing modules. Additionally, practical tests were conducted to assess the model's practicability. Comparing the model with other network threat management methods, the following conclusions were drawn:

Firstly, the system utilizes honeypot technology to achieve active defense measures for the network environment while withstanding continuous attacks, demonstrating that the system has achieved its security and seduction objectives.

Secondly, the system employs numerous high-interaction honeypots as information collection tools, providing better seduction capabilities than low-interaction honeypots and yielding more comprehensive information.

Thirdly, the system formats the collected data, utilizing the attacker's IP address as the primary key for classification and assessing the harm caused by the attacker to score and grade it. Users can quickly understand the attacker's operation and other relevant information through the attacker's IP address.

Fourthly, the system offers a user-friendly web interface, reducing the threshold for usage and promoting the more widespread application of honeypot technology by simplifying the user's operational difficulty and enhancing their experience.

Finally, the system adapts to a variety of operating systems, matches the network environment closely, and has significant improvements in security and proactivity when compared to traditional network threat management systems.

**Author Contributions:** Conceptualization, J.Y.; methodology, J.Y.; software, X.Y. and H.Z.; validation, X.Y., H.Y. and Y.K.; formal analysis, J.Z. and J.Y.; investigation, X.Y.; resources, J.Y.; data curation, H.Z.; writing—original draft preparation, X.Y.; writing—review and editing, J.Y.; visualization, Y.K.; supervision, J.Y.; project administration, J.Y.; funding acquisition, J.Y. All authors have read and agreed to the published version of the manuscript.

**Funding:** This  work is supported by the National Natural Science Foundation of China under Grant62002028 and Research Innovation Fund for College Students of Beijing University of Posts and Telecommunications.

**Informed Consent Statement:** Informed consent was obtained from all subjects involved in the study. Written informed consent has been obtained from the patients to publish this paper.

**Data Availability Statement:** The data underlying this article are available in the article.

**Acknowledgments:** The authors express great appreciation to Guoqiang Xing.

**Conflicts of Interest:** The authors declare no conflict of interest.

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
