# Peer review of "A Highly Interactive Honeypot-Based Approach to Network Threat Management"

_futureinternet, doi:10.3390/fi15040127_

Round 1

Reviewer 1 Report

The paper needs a careful review of the English writing. Several sentences are not clear, sometimes there are repetitions etc.

The focus of the manuscript is not clear. It is a mix of literature review on honeynets and research of innovative solutions. Nonetheless the technical content related to the research part is too generic. The details on the solution proposed by the authors are minimal. To be acceptable the authors should provide more details about the solution they have developed and some more operational example of its use.

In Section 5 for instance they claim that the advantage of their solution is that it collects more data, but fail to explain in some detail how they are used, which kind of information are summarized from those data etc.

Author Response

Dear Reviewer,

I am writing to express my sincere gratitude for your invaluable feedback on my manuscript. Your insightful comments and suggestions have helped me improve the overall quality of my paper.

I am pleased to inform you that I have made extensive revisions to the manuscript, focusing specifically on strengthening the English logic and improving the clarity of my writing. Additionally, I have added technical details to Chapters Three, Four, and Five, as well as restructured the manuscript to enhance its overall coherence.

I believe these changes have significantly improved the quality of my paper, and I hope that they will meet your expectations. Once again, I would like to thank you for your time and effort in reviewing my manuscript. Your feedback has been truly invaluable to me.

Sincerely,

Xingyuan Yang

Reviewer 2 Report

The work concerns the field of data and equipment security and aims to improve them by using an active defence solution based on honeypot. The authors identify the problems of the traditional use of honeypots (with a low degree of interactivity – Section 2) and propose a solution to improve the protection of equipment and increase information security, that imply the use of interactive and modular honeypot technology for active defence.

For increasing the defence efficiency, the authors propose the use of highly interactive honeypot technology (whose framework is presented in Figure 1) that lures attackers into predefined sandboxes to observe their behaviour and attack methods, but also performs a variety of advanced functions such as network threat analysis, virtualization, vulnerability perception, trace enhancement and camouflage detection.

Through modularization, the honeypot environment is differentiated from the central data processing node, through which honeypots are managed; thus, adding, deleting, modifying honeypots and other operations are easy to do, making the whole activity of working with them easier.

In fact, the central node is the core module of the entire structure. It implements data storage, deploys multiple honeypots, controls connections, and completes a series of data processing functions: data storage, integration into the structure of honeypots, connection control and completion of a series of data processing functions. In addition, based on the various operations performed at the level of honeypots, data collection, organization, summarization and analysis of data, event reports, formation of attacker profiles for visualization and easy use of attack information templates are done here.

After the presentation of the realized survey of the existing network threat management methods in Section 2, Section 3 is dedicated to the presentation of the Central Node Model In the High-Interactivity Honeypot System (whose framework is presented in Figure 1), together with the Information Capture system (presented in section 3.1 - Figure 2), the connection Control Module (presented in section 3.2 – Figure 3 and Figure 4), the Honeypot Deployment (presented in section 3.3 – Figure 5), Data analysis and processing (presented in section 3.4 – Figure 6) and the Data storage solution (presented in section 3.5)

Section 4 is dedicated for Implementation and results of analysis of high Interactivity honeypot system. In this section the authors present a Comparison of interactive honeypots (Figure 7), the Central node interaction model, the Central node data flow model, the implementation for the High interaction honeypot (with a schematic diagram of the attack – Figure 10) and the overall structure design implementation. 

Many experimental comparisons were made, and it was proved that the proposed method has significant advantages compared to the traditional honeypot technology.

Weaknesses

       A brief description of the structure of the paper is required.

       A brief description of the research methodology and the tools used in evaluation.

       The name for figure 4 is redundant.

       A Conclusion section is necessary (or maybe section 6 is considered as a conclusions section …)

Author Response

Dear Reviewer,

Thank you for your valuable feedback on my article. I appreciate your time and effort in evaluating my manuscript and providing insightful comments.

I am writing to inform you that I have extensively revised my article based on your feedback. Specifically, I have strengthened the English logic and expression throughout the manuscript, added an introduction section to provide a better overview of the article structure, supplemented technical details in Chapter 3, 4, and 5, corrected the label of Figure 4, and included a conclusion section to summarize the findings.

I am grateful for your guidance and constructive criticism, which have helped me improve the quality of my work. Please let me know if there are any further suggestions or concerns that you would like me to address.

Once again, thank you for your valuable feedback and your time in reviewing my manuscript.

Best regards,
Xingyuan Yang